# Cell States and Interactions of CD8 T Cells and Disease-Enriched Microglia in Human Brains with Alzheimer’s Disease

**DOI:** 10.3390/biomedicines12020308

**Published:** 2024-01-25

**Authors:** Mai Yamakawa, Jessica E. Rexach

**Affiliations:** Department of Neurology, University of California Los Angeles, Los Angeles, CA 90095, USA; myamakawa@mednet.ucla.edu

**Keywords:** neuroinflammatiosn, microglia, T lymphocytes

## Abstract

Alzheimer’s disease (AD) is a multi-stage neurodegenerative disorder characterized by beta-amyloid accumulation, hyperphosphorylated Tau deposits, neurodegeneration, neuroinflammation, and cognitive impairment. Recent studies implicate CD8 T cells as neuroimmune responders to the accumulation of AD pathology in the brain and potential contributors to toxic neuroinflammation. However, more evidence is needed to understand lymphocytes in disease, including their functional states, molecular mediators, and interacting cell types in diseased brain tissue. The scarcity of lymphocytes in brain tissue samples has limited the unbiased profiling of disease-associated cell types, cell states, drug targets, and relationships to common AD genetic risk variants based on transcriptomic analyses. However, using recent large-scale, high-quality single-nuclear sequencing datasets from over 84 Alzheimer’s disease and control cases, we leverage single-nuclear RNAseq data from 800 lymphocytes collected from 70 individuals to complete unbiased molecular profiling. We demonstrate that effector memory CD8 T cells are the major lymphocyte subclass enriched in the brain tissues of individuals with AD dementia. We define disease-enriched interactions involving CD8 T cells and multiple brain cell subclasses including two distinct microglial disease states that correlate, respectively, to beta-amyloid and tau pathology. We find that beta-amyloid-associated microglia are a major hub of multicellular cross-talk gained in disease, including interactions involving both vulnerable neuronal subtypes and CD8 T cells. We reproduce prior reports that amyloid-response microglia are depleted in *APOE4* carriers. Overall, these human-based studies provide additional support for the potential relevance of effector memory CD8 T cells as a lymphocyte population of interest in AD dementia and provide new candidate interacting partners and drug targets for further functional study.

## 1. Introduction

Alzheimer’s disease (AD) is the most common cause of late-onset dementia, and it is characterized neuropathologically by the co-occurrence of senile plaques containing beta-amyloid (Aβ) and neurofibrillary tangles composed of hyperphosphorylated tau (pTau) [1]. Beta-amyloid accumulation is known to precede pTau accumulation and cognitive impairment by ~10 years based on longitudinal biomarker studies, indicating that AD is a multi-step disorder spanning over 10 years [2]. Recent advances in genetics highlight neuroinflammation as a major causal factor in AD pathogenesis with the identification of multiple immune-related AD risk genes such as *CD33*, *TREM2*, and major histocompatibility complex (MHC) genes [3,4,5,6].

The role of neuroinflammation in the pathogenesis of AD is a major area of investigation [7,8,9,10,11]. Single-nuclear RNA sequencing (snRNAseq) studies of human AD brain tissue support a greater diversity of disease-associated states involving immune cells and associated pathways in microglia that only partially overlap the disease-associated microglia (DAM) pattern first described in AD mouse models [12,13,14,15]. Corresponding with diverse microglia transcriptomic states, microglia play varied roles in disease, ranging from beneficial functions like clearing toxic Alzheimer’s pathology and dead cells to maladaptive roles such as synapse clearance, triggering inflammatory cascades, spreading pathology and driving cellular injury [13,16,17,18,19,20]. Among them, microglial responses against Beta-amyloid and senile plaques have been particularly well characterized both in AD mouse models and human postmortem brain to delineate associated transcriptomic cell states and marker genes [21,22]. For example, Nguyen et al. validated markers of three distinct microglial populations whose abundance vary based on the co-existence of *APOE* or *TREM2* risk variants: dystrophic microglia (marked by *FTL* and *FTH1*), amyloid-responsive microglia (ARM) expressing *CD163*, *BIN1*, *MSA4A6A*, and homeostatic or motile microglia [9]. Among them, amyloid-responsive microglia (ARM) were depleted in cases carrying the risk-associated *APOE4* alleles [9]. Unlike microglial response and neuroinflammation against Beta-amyloid accumulation, late neuroinflammation involving pTau is less well characterized. Additionally, the topic of cell–cell communication involving microglia with other cell types, including immune cells, is an emerging area of investigation, but it also remains poorly understood in the context of neurodegenerative disease [23]. Recognizing the significance of these interactions is crucial, as they broaden the potential impact of targeting microglial genes and pathways for therapeutic modulation.

The single-cell sequencing of immune cells in human spinal fluid has revealed significant changes in CD8 T lymphocytes in AD cases compared to controls [24]. Recent functional studies support the causal involvement of CD8 T cells in late neuroinflammation involving pTau in mouse models and organoid models [15,25,26]. Chen et al. demonstrated a causal role of CD8 T cells in tau-mediated neurodegeneration and brain atrophy in mice models expressing a human tau mutation (P301S *MAPT*) [25]. T cell infiltration and the induction of interferon-γ and neuroinflammatory pathways in glia have been demonstrated in a human 3D organoid system [26]. However, knowledge of transcriptomic signatures of CD8 T cells in human AD brains is lacking partly due to the rarity of CD8 T cells in the brain, posing technical challenges to their characterization. Furthermore, while both microglia and T cells emerge as important players in neuroinflammation and AD pathogenesis, the scope of interaction between the two cell types in human AD brains remains to be fully characterized.

Here, we analyzed a large snRNAseq dataset of the middle temporal gyrus from 84 donors with AD or unaffected control from the Seattle Alzheimer’s Disease Atlas (SEA-AD) cohort [27] to characterize disease-associated microglia and CD8 T cells and their interactions. We depict gene co-expression network modules that correlate with amyloid-responsive, dystrophic, and pathology-associated microglia [9,10]. We demonstrate that CD8 T cells are increased in AD brain but not natural killer (NK) cells or CD4 T cells. Using cell–cell interaction (CCI) analysis, we describe ligand–receptor pairs (LR pairs) unique to CD8 T cells and disease-enriched microglia.

## 2. Materials and Methods

### 2.1. Data and Samples

Single-nuclear RNA sequencing data of the middle temporal gyrus from 84 donors with AD (n = 42) or healthy control (n = 42) were downloaded from the Seattle Alzheimer’s Disease Atlas (SEA-AD) for microglia and perivascular macrophages (Microglia-PVM), layer 2/3 intratelencephalic excitatory neurons (L2/3 IT), layer 4 intratelencephalic excitatory neurons (L4 IT), parvalbumin interneurons (Pvalb), somatostatin interneurons (Sst), astrocytes, oligodendrocytes, and oligodendrocytic progenitor cells (OPC). All SEA-AD data used in this study including detailed sequencing methods are accessible at the SEA-AD Brain Cell Atlas (https://portal.brain-map.org/explore/seattle-alzheimers-disease, (accessed on 23 February 2023)). Metadata provided with the SEA-AD dataset include case ID, age at death, sex, postmortem interval (PMI), pathological stages (ADNC, Thal phase, CERAD score, Braak stage), *APOE4* carrier status, race, and ethnicity. These data were curated and made public by the Allen Brain Institute as described by Gabitto et al. [27]. As per Gabitto et al., Lewy body disease scores were based on the National Institute on Aging guidelines [27,28,29].

The analysis was performed with Seurat (version 4.4.0) in R (version 4.3.0). Samples were removed from the microglial population to reduce confounding and to limit poorly represented neuropathological traits. The implemented filters aimed to refine the sample set, ensuring a focused analysis of changes in microglia cells related to AD. Specifically, we removed all cases with diffuse neocortical Lewy body pathology (high “LBD”), as they were disproportionately identified in cases over 90 years of age and only found in cases with AD co-pathology (Appendix A). In addition, we removed 4 cases that were the only cases with a certain Braak stage among normal or dementia cases. For the CD247+ lymphocyte population, due to their scarcity, we did not filter out cases due to age or Braak stage. However, upon clustering of the lymphocyte population, one donor comprised one independent cluster (donor H20.33.004), and this donor was removed from the downstream analysis. Based on these filters alone, we removed cases with advanced AD pathology but no cognitive impairment, as well as cases that were clinically diagnosed as AD but had minimal AD pathology. This filtering reduced the total number of Microglia-PVM analyzed from 39,200 cells from 84 donors (42 controls, 42 AD) to 11,052 cells from 28 donors (15 controls, 13 AD).

### 2.2. Annotation of Lymphocyte Subclasses and States

To classify representative lymphocytes and microglial states in AD and unaffected controls (Schema: Figure 1), we extracted all cells in the SEA-AD dataset that were designated as the group containing microglia or perivascular macrophages (“Micro-PVM”) based on inference-based cell type annotation, as described [27,30]. To further annotate lymphocytes including T lymphocytes and natural killer (NK) cells, we used the marker gene *CD247* (T cell receptor ζ chain). As expected, *CD247* expression was low in all other cell clusters in SEA-AD (Figure 2A). Furthermore, consistent with identifying lymphocytes, the top 10 marker genes discriminating the CD247+ subcluster from others included *THEMIS*, *CD247*, and *CD96*. In order to identify subtypes of lymphocytes among the *CD247*+ subcluster, we isolated these cells and performed another round of clustering using Seurat. This identified 7 distinct subclusters. Lymphocyte subclusters were then annotated into their subtypes and states using markers manually curated from the literature to represent distinct types of tissue-resident lymphocytes (Appendix A), such as *NCAM1* for NK cells, *CD8A/B* for CD8 T cells, and *PRF1* and *GZMA* for cytotoxic cells [31,32,33,34,35].

### 2.3. Lymphocytes Cell Proportion and Differential Gene Expression Analysis

We used multivariable linear regression to calculate the proportions per donor of CD8 T cells, NK cells, and CD4 cells relative to all “Microglia-PVM” designated cells, including biological and technical covariables (age, sex, and PMI) using lme4 (version 1.1-35.1), lmerTest (version 3.1-3), and modelssummary (version 1.4.3) packages. Similarly, we applied multivariable linear regression to define relationships between T cell counts and either donor cognitive status or AD neuropathological traits, with each run as separate models, using the same series of biological and technical factors listed above (see sample metadata in Appendix A). In order to measure differential gene expression among CD8 T cells, we filtered out genes whose maximum expression was less than 200 counts per million across all cells (7140 genes remained) and then performed differential gene expression analysis using linear regression to identify genes differentially up and downregulated in CD8 T cell from cases with AD dementias vs. unaffected controls (see Appendix A for additional details) [36].

### 2.4. Weighted Gene Co-Expression Network Analysis of Lymphocytes and Microglia

Toward the goal of defining relationships between lymphocyte and microglia signaling states or functional classes, we leveraged weighted gene expression analysis to define stable gene sets (“modules”) that may relate to changes in biological signaling or function among lymphocytes or microglia in disease. We selected this approach because markers of microglia disease states based on differential gene expression alone have been reported to be highly variable across AD studies [21], and we reasoned that the modules based on WGCNA, which are highly reproducible [15,22], would offer alternative gene markers based on robust correlations with the biological phenomenon that are altered in disease. Therefore, we applied high dimension-Weighted Gene Co-expression Network Analysis (hdWGCNA; version 0.2.18), which is a version of WGCNA adapted for single-cell datasets (WGCNA: version 1.72-1) [37]. In this analysis, gene sets known as “modules” are generated, and their combined expression is represented as a module eigengene (ME). By quantifying the expression of the ME in cells or samples, we measure the differential expression of the biological changes that are represented by the module. We performed this analysis based on published code without modification [37]. Details regarding the application of this analysis in this study, including module generation parameters, are described in the Appendix A. We performed gene ontology and pathway enrichment analysis using the enrichR package (version 3.2) [38,39,40]. We generated ten microglia (mg) modules that were designated mg1–10, ranging from the largest (1) to the smallest (10) module. In addition, we generated 19 CD8 T cell modules that were designated CD8T1-19.

To further annotate microglial modules with respect to substantial precedent work, we curated, published, and validated markers from the literature and tested module enrichment using Fisher’s exact test. We prioritized markers from human studies with distinct relationships to AD neuropathological traits. This included microglia differentially associated with beta-amyloid, *APOE* and *TREM2* genotypes validated in human brain tissue by immunohistochemistry in Nguyen et al. [9], including homeostatic microglia (*TMEM119*), motile microglia (*ARHGAP15*), dystrophic microglia (*FTL*, *FTH1*), and amyloid-responsive microglia (ARM; *CD163*) [5]. In addition, we leveraged microglial gene sets reported from human AD brain tissue to be differentially expressed in relationship to beta-amyloid, including *CD163* in addition to *APOE* as well as distinct genes related to pathological tau load, including *LRRK2* and *GPNMB* [10]. Finally, we used commonly used markers of homeostatic microglia, including *TMEM119* and *CX3CR1* [41]. Based on these markers, we annotated the four microglial modules significantly associated with disease status as representing signatures of either amyloid-related (ARM; mg1), pathology-associated (mg3), dystrophic (mg2), or homeostatic (mg8 and mg10) (Appendix A). We identified subsets of cells representatives of each module (for example ARM microglia) by designating the cell with the highest module eigengene (ME) expression (top 20% by rank) unique to a particular module (Appendix A). In addition, we identified a subset of cells with ME scores among the top 20% for both mg1 and mg3 modules. Building on the understanding that beta-amyloid precedes tau pathology chronologically, we designated this subset of microglia with intermediate scores for both ARM and pathology-associated modules as putative intermediate phase or “transition” microglia (Appendix A). These cells were then used for downstream cell-based analyses, including cell–cell interaction analysis.

### 2.5. Cell–Cell Interaction Analysis

We used the package Multinichenetr (version 1.0.0) to perform cell–cell interaction analysis to compare interactions involving different types of microglia and lymphocytes with other brain cell types. In addition to interactions involving glia, we were interested specifically in distinguishing interactions involving subclasses of neurons depleted from disease cases as a function of cognitive decline and neuropathology [27]. For this reason, we chose to use Multinichenetr for CCI analyses, which measures differential CCI across cell populations and sample groups [42]. We performed CCI as described in the Multinichenetr package without modification, and we included only paracrine interactions [42]. We applied the analysis to a subset of 23 cases in which both disease-associated microglia and CD8 T cell populations were represented. We then performed pathway enrichment using enrichR and KEGG 2021 Human and Wikipathway 2023 Human databases, which we applied to the top 100 ligands per sender cell type, top 100 receptors per receiver cell type, or top 100 ligand-receptor pairs per specific sender and receiver pairs. To further annotate ligands and receptors involving CD8 T cells according to pathways represented by hdWGCNA modules for CD8 T cells, we calculated and tabulated gene-module connectivity scores (kME) for each ligand or receptor against each CD8 T cell module (Appendix A).

## 3. Results

### 3.1. Characterization of Brain-Tissue-Associated Lymphocytes

#### 3.1.1. Differential Composition Analysis and Marker-Based Annotation Demonstrate an Increase in Effector Memory CD8 T Cells in Samples with AD Dementia

Studies of human cerebral spinal fluid and blood have demonstrated various compositional changes involving peripheral lymphocytes in Alzheimer’s disease (AD) dementia [24,43]; however, the specific alterations of brain lymphocytes within AD brain tissue remain undefined. To fill this gap, we performed snRNAseq-based cell classification and compositional analysis to characterize peripheral blood cells present in brain tissue of AD dementia and matched unaffected control cases. We classified as lymphocytes a specific subcluster of 800 cells from 84 cases in the SEA-AD dataset, based on its unique enrichment for the lymphocyte marker gene *CD247*, which encodes the T cell receptor subunit CD3ζ (Figure 2A). To annotate specific subtypes of *CD247*+ lymphocytes, we performed subclustering and marker-based annotation [31,32,33,34,35]. We identified seven subclusters that were uniquely enriched for markers of either natural killer (NK) cells, CD4 T cells, and CD8 T cells (Figure 2B; see Section 2). Among these, CD8 T cells were the most abundant lymphocytes in the dataset (384 cells from 70 donors). The expression of specific marker genes can distinguish different activation and maturation stages of CD8 T cells (Appendix A) [31,32,33,34,35]. Among these, we observed markers for effector memory T cells (TEMRA) including high CD45RA positivity concurrent with slightly increased cytotoxic enzyme genes (*PRF1*, *GZMA*, *GZMB*, and *NKG7*; Figure 2C, Appendix A). Therefore, based on the snRNAseq of human brain tissue, we identified various populations of CD3ζ+ lymphocytes, defined their empiric transcriptomic signatures (presented in Figure 2C and Appendix A), and classified subtypes based on marker gene expression, including TEMRA as the most abundant CD3ζ+ cell population.

To assess the relative abundance of CD8 T lymphocytes in cases with AD dementia vs. unaffected controls, we next performed cell type compositional analysis, applying multivariable linear regression to control for technical and biological covariates including age, sex, and PMI (see Section 2). We found that CD8 T cells were significantly enriched in AD dementia brain samples (Figure 2D; log2FC = 0.21, *p*-value = 0.036, n = 42 AD and 42 control cases). In contrast, we did not observe significant increases in the expression of other lymphocyte classes, including NK cells and CD4 T cells (Appendix A). There was no difference in CD8 T cell proportions between female and male (female versus male sex: log2FC = 0.087, *p*-value = 0.41; 54 female, 35 male; Wilcoxon rank sum test). We also found no significant difference in CD8 T cell proportions relative to *APOE4* carrier status in cases with dementia, but we found a positive trend in *APOE4* carriers without dementia that was not statistically significant (dementia: log2FC = 0.07, *p*-value = 0.69, n = 16 *APOE4* carriers and 26 *APOE4* non-carriers; cognitively normal: log2FC = 0.74, *p*-value = 0.07, n = 9 *APOE4* carriers and 33 *APOE4* non-carriers, Wilcoxon rank-sum test). To determine if the CD8 T cell proportion was further associated with AD-specific neuropathology, we examined CD8 T cell abundance relative to stages of beta-amyloid (Thal), hyperphosphorylated tau (Braak), or neuritic plaque density (CERAD score). While we observed positive correlations between CD8 T cell proportion and each of these scores among dementia cases, they were not statistically significant (Figure 2D). Therefore, using snRNAseq compositional analysis, we observed an enrichment of CD8 T lymphocytes bearing features of TEMRA as a feature of AD dementia brain tissue with AD pathology that was independent of the *APOE4* genotype.

Having assessed the transcriptome of brain-associated CD8+ lymphocytes from AD and control cases, we next addressed whether we could detect differentially expressed genes based on snRNAseq. Analyzing 7140 genes that had sufficient detection in CD8 T cells over 70 donors for differential gene expression analysis, we identified 13 genes with significant differential expression in disease samples based on significance threshold (abs(log2FC) > 1, FDR corrected *p*-value < 0.05; Appendix A). Prominent among these genes (12 of 13) were mitochondrial genes and non-coding RNAs, which could indicate damaged cells or ambient RNA (Appendix A) [44,45]. In addition, CD8 T lymphocytes in disease samples had higher expression of *ERC1* (log2FC = 1.02, FDR-corrected *p*-value = 0.026; limma), which is a regulatory subunit of *IKKB* believed to recruit the IkappaB/NFKBIA complex [46]. Therefore, while underpowered for a full transcriptomic analysis, these data indicate a potential for the differential regulation and/or activity of CD8 T cells present in AD brain tissue samples.

#### 3.1.2. Weighted Gene Co-Expression Network Analysis of CD8 T Cells Reflects Terminal Differentiation and Multiple Signaling States

To capture groups of gene sets (network modules) whose expression changes are correlated to each other and may be related to underlying changes in cell state or signaling pathways, we next performed weighted gene co-expression network analysis (WGCNA) on CD8 T cells (see Section 2 [37,47]). This yielded 19 network modules, among which two (CD8T2 and CD8T4) were more highly expressed in cases with dementia compared to cognitively normal controls (CD8T2: log2FC of module eigengenes = 2.44, FDR-corrected *p*-value = 0.0352; CD8T4 log2FC = 1.63, FDR-corrected *p*-value = 0.0352; linear regression with mixed random effect model as described in Appendix A) (Figure 3, Appendix A). Therefore, we annotated the hub genes and associated pathways for CD8T2 (504 genes) and CD8T4 (743 genes) to consider biological pathways and cell states associated with CD8 T cells in AD dementia samples (see Appendix A for GO term enrichment). Genes more strongly connected to CD8T2 (kME, see Section 2) included effectors of lymphocyte cytotoxicity including *GZMK* (granzyme K) and *PRF1* (perforin 1) together with mitochondrial and ribosomal genes (Figure 3B,C; Appendix A) [34]. In contrast, among the top-ranked “hub” genes of the alternative CD8T4 module were modifiers of inflammation or immune signaling, or “immunomodulatory” genes (Figure 3D,E), including *IL7R*, which is crucial for the survival of effector memory T cells and naïve CD8 T cells [48,49,50], *CD226*, a regulator of lymphocyte proliferation and cytokine production [51], and *TIGIT*, an immune checkpoint regulator [52]. Additional CD8T4 genes included killer cell lectin family genes, which are known to be expressed in subsets of TEMRA such as virtual memory T cells as well as NK cells [53,54]. Therefore, we identified two modules that represent distinct facets of T cell signaling, encompassing cytokine-mediated immunomodulation (CD8T4) and cytotoxic granule formation and cell killing (CD8T2), which are overall consistent with the CD8 T cells we identified being mainly terminally differentiated effector memory T cells.

### 3.2. Characterization of Disease-Associated Microglia

#### Weighted Gene Co-Expression Network Analysis of Microglia Defines Distinct Beta-Amyloid Responsive and Pathology-Associated Microglia That Reproduce Validated Markers and Disease Traits

Microglia are brain resident immune cells previously implicated in transgenic models to be major mediators of lymphocyte cross-talk in the context of AD pathology [25,26]. However, relationships between microglia and lymphocytes remain poorly defined in the human disease brain, including whether they involve particular states of microglia, such as microglia associated with beta-amyloid plaque and *APOE4* status or microglia more associated with tau pathology and neurodegeneration [9,13,16,17,18,19]. Therefore, we next utilized the SEA-AD dataset to define populations of microglia associated with different types of AD pathology and/or *APOE* genotype. To define reproducible states and associated marker genes, we applied WGCNA to represent microglia transcriptomic profiles as gene co-expression modules to define robust markers and stable gene sets for pathway analysis. For this analysis, we selected a subset of cases from the SEA-AD dataset with typical AD pathology and dementia, removing cases with co-pathology or poorly represented neuropathological traits (as described in Section 2; n = 28 cases, 11,052 microglial cells remaining). This analysis defined 10 microglial modules, which we named mg1–10 (Appendix A). Two of these modules were significantly enriched in AD dementia samples (mg1 and mg3; FDR < 0.1, linear regression with mixed random effect model as described in Appendix A). Hub genes distinctly marking these modules included genes previously reported in disease-associated microglia including *CD163* (mg1; Figure 4A–C), a validated marker of amyloid responsive microglia (ARM) [9], and *GPNMB* (mg3; Figure 4D), a microglial gene correlated to tau pathology load in prior studies [10]. We next performed gene ontology analysis of the disease-associated modules mg1 and mg3 modules to characterize represented biological pathways (Appendix A). The module mg1 was enriched for neutrophil-mediated immunity, positive regulation of transcription from RNA polymerase II promoter in response to hypoxia, complement cascades, and cholesterol metabolism, whereas the mg3 module was enriched for lysosomal structures and signaling genes (Appendix A). In contrast, genes associated with antigen processing and presentation were significantly enriched in the module mg2, including *HLA-C*, *HLA-DRA*/*DRB1*, and *TAP1* (adjusted *p*-value = 0.005; Appendix A). Therefore, using WGCNA, we identified several distinct microglial modules bearing reproducible marker genes and representing distinct biological pathways.

Multiple traits defined in prior work characterize amyloid-responsive microglia (ARM), including (1) the marker gene *CD163*, (2) positive association with beta-amyloid stage, and (3) relative depletion in *APOE4* carriers compared to non-carriers [5]. To ascertain if the mg1 module in SEA-AD represents ARM, we tested these identical associations in the SEA-AD dataset. In line with previous findings, we observed diminished mg1 expression in *APOE4* carriers compared to non-carriers in individuals with AD dementia and high beta-amyloid burden (*APOE4* carriers: log2FC of module eigengenes for high vs. low beta-amyloid burden = −2.02, *p*-value = 0.17; *APOE4* non-carriers: log2FC = 1.94, *p*-value = 0.00076; across 11,052 cells from 28 donors; linear regression with mixed random effect model as described in Appendix A) (Figure 4B). We also observed a significant increase in mg1 module expression from the early stages of senile plaque burden (CERAD score) but not with neurofibrillary tangle burden (Braak stages) until Braak stage VI (Figure 4C). Hence, the characteristics of mg1 align with the definition of ARM according to Nguyen et al. [5]. Furthermore, supporting its association with beta-amyloid, mg1 showed an enrichment with differentially expressed genes for beta-amyloid identified by Smith et al. [6] (Appendix A). Therefore, based on this evidence, we adopted the prior designation and labeled mg1 as the ARM module.

In contrast to mg1, genes within mg3 have previously been associated with phosphorylated Tau in human Alzheimer’s disease (*AD*) brain samples [6] (Appendix A; see Section 2). Correspondingly, mg3 expression increased significantly with advancing AD neuropathological stages in dementia cases, including tau pathology (Braak; Figure 4C). The module mg3 did not exhibit a significant correlation with *APOE4* carriers either with or without dementia (Appendix A). Consequently, we designated mg3 as the “pathology-associated” module. Similarly, using reproducible and validated markers, we annotated additional modules including homeostatic (mg8 and mg10; *TMEM119*+*CX3CR1*+) and dystrophic (mg2; *FLT*+) (see Section 2; Appendix A). Derived from weighted gene co-expression network analysis (WGCNA), this robust annotation offered modules with specific associations to Alzheimer’s disease (AD) pathology and *APOE* genotype, facilitating the comparison of immune-related cell–cell signaling across microglia disease states in the AD brain.

### 3.3. Cell–Cell Interaction Analysis

#### 3.3.1. MultiNicheNet Reveals Markedly Increased CCI Involving Depleted Neurons and Beta-Amyloid Responsive Microglia in AD Dementia Brain Samples

To elucidate potential functional consequences of the identified disease-associated immune cell types and states, we conducted cell–cell interaction analysis to compare interactions gained or lost in the diseased brain. While we applied the analysis broadly across all major glial cell types and multiple neuronal subclasses, we focused the analysis specifically on immune cells, pathology-associated microglia, and depleted vs. compositionally spared neuronal subclasses. To achieve this, we employed the recently developed cell–cell interaction (CCI) analysis tool, MultiNicheNet, to enable differential CCI analysis across cell types and disease conditions based on ranked scores.

Our analysis resulted in 359,376 ligand–receptor (LR) pairs scored and differentially active in disease vs. control conditions, which were further ranked for each cell type [42]. As a preliminary test case, we first compared interactions involving neuronal subclasses that are either depleted or compositionally spared in dementia cases in the SEA-AD dataset, reasoning we would expect to find regulators of cell death, stress, and/or survival ranked more highly among LR pairs targeting depleted vs. spared neurons. Specifically, in the SEA-AD dataset, including in samples used for this analysis (Appendix A), the disease-depleted neurons include layer 2/3 intratelencephalic excitatory neurons (L2/3 IT) and somatostatin inhibitory neurons (Sst), which we will refer to as vulnerable neuronal population [42]. Overall, dementia samples contributed to more of the highly ranked LR pairs compared to controls, contributing 39 out of the top 50 most highly ranked LR pairs defined by the analysis (Figure 5A and Appendix A) [42,55]. Moreover, many highly ranked LR pairs that were gained in disease samples compared to controls involved one of the two “vulnerable” neuronal types (L2/3 IT or Sst) [27], supporting the use of this analysis to identify disease-associated CCI (Appendix A). Furthermore, disease-specific LR pairs with L2/3 IT and Sst neurons were enriched for pathways of neurodegeneration including Wnt signaling and *LRP6* [56,57]. In contrast, these same pathways were not enriched among LR pairs involving neuronal subtypes that were compositionally spared (L4 IT and Pvalb). Thus, CCI analysis identified LR pairs with differential activity in disease and control cases, emphasizing neurodegeneration-related LR pairs specifically enriched among neurons selectively depleted from disease tissues.

Next, we leveraged CCI analysis to assess whether different microglia states, defined by WGCNA modules, involved shared or distinct disease-associated cell–cell interactions. To enable this, we first identified subpopulations of microglia representative of each microglia module, defining ARM, pathology-associated, transition, dystrophic, and homeostatic microglia (see Section 2). We then compared CCIs across microglia in these different disease states, focusing on paracrine interactions either upregulated or downregulated in disease compared to control cases (see Section 2). Notably, interactions involving ARM dominated the LR pairs gained in disease, demonstrating over 10 times as many interactions as the next most involved microglial state (Figure 5B). Many of the LR pairs unique to ARM involved neuroprotective and glioprotective factors, such as *APOE-SORL1* [58], *CD47*-*SIRPA* [59], and *PSAP-GPR37L1* [60]) (Appendix A). In contrast, disease-enriched LR pairs involving pathology-associated microglia included mediators of cellular damage, including complement (*C3-CD46*) [61], cholesterol scavenger receptors (*PTDSS1-SCARB1*) [62], and drivers of microglia-mediated neuronal death (*CNTN4-PTPRG)* [63] (Appendix A). Other LR pairs were shared by both types of disease-associated microglia such as interactions involving the AD disease gene *APOE* with *LRP5*, but these were different from those with homeostatic or dystrophic microglia (Appendix A). These findings indicate that microglial states associated with either beta-amyloid, *APOE* status, and/or tau pathology may participate in distinct cell–cell interactions, and they highlight broad and abundant disease-associated multicellular engagement with amyloid-responsive microglia (ARM).

#### 3.3.2. Disease-Associated CCIs Involving CD8 T Cells

Next, we leveraged CCI to examine interactions between brain resident cells and brain tissue-associated CD8 T lymphocytes to annotate related functional pathways and candidate mediators of neuroimmune signaling changes in AD dementia brain tissue. By applying MultiNicheNet (see Section 2), we identified 11,366 differential CCIs specific to dementia samples involving CD8 T cells [42]. LR pairs with CD8 T cells that were gained in the disease prominently included cell adhesion molecules, growth factors, and receptor tyrosine phosphatases (Appendix A). The majority of CCIs involving CD8 T cells were with microglia, and disease-associated microglia had the majority of CCIs with CD8 T cells (ARM > transition > pathology-associated). In contrast, CCIs between CD8 T cells and either homeostatic or dystrophic microglia were absent in disease tissues and only present in unaffected controls (Figure 6). Therefore, interactions between CD8 T cells and microglia shifted markedly from normal to dementia conditions with interactions predominantly gained with disease-enriched microglia and lost with homeostatic/dystrophic microglia.

We further annotated CCIs involving CD8 T cells with their functional pathways using the predetermined CD8 T cell network modules, including CD8T2 (cytotoxic) or CD8T4 (immunomodulatory) modules. The majority of disease-associated CD8 T interactions involved genes in the immunomodulatory module, including cell–cell adhesion interactions *FLRT2-UNC5A-D* and *NRG1-ITGAV/ERBB4* (L2/3 IT and Sst as interacting partners) (Appendix A). In contrast, CCI involving genes within the CD8T2 cytotoxicity module were dominated by *ADAM10*, an AD GWAS gene and metalloprotease with many target genes including *APP* [64]. CCI with *ADAM10* included multiple LR pairs that varied by target cell type, including *NOTCH2* (with ARM), *CD44* (with pathology-associated microglia), and TSPANs (L2/3 IT and Sst). Having observed the differential composition of ARM to be a consistent feature of *APOE4* carriers, we examined if the *NOTCH2* expression differs between *APOE4* carriers and non-carriers in dementia cases. We found *NOTCH2* to be significantly upregulated in ARM from *APOE4* carriers compared to those from *APOE4* non-carriers (estimate = 0.12, *p*-value = 0.0198, adjusted *p*-value = 0.099; linear regression with mixed random effect models) (Appendix A). Hence, our analysis pinpointed *NOTCH2* as a potential mediator in cytotoxic CD8 T cell interactions with amyloid-responsive microglia (ARM). Moreover, *NOTCH2* was found to be upregulated in *APOE4* carriers where ARM depletion occurs. This emphasizes that cell–cell interactions (CCIs) involving CD8 T lymphocytes play diverse roles, evident in both ligand–receptor (LR) pairs and functional pathways, highlighting ARM as a potential interacting microglia type due to significant gains in disease-associated CCIs.

## 4. Discussion

To achieve a systematic view of brain-tissue associated lymphocytes and their relationship with microglial disease states, we analyzed the large snRNAseq dataset of 1,240,908 cells from 84 AD and healthy control donors generated from the Seattle Alzheimer’s Disease (SEA-AD) cohort [27]. To address the significant need for human-brain-based evidence of neuroimmune targets for functional and mechanistic study, we focused our analysis on rare brain tissue-associated lymphocytes and microglia, which are the resident immune cells of the brain. Our analysis demonstrated a significant increase in CD8 T cell abundance in the brains of individuals with dementia due to Alzheimer’s disease compared to individuals without dementia. AD brain-enriched lymphocytes matched with effector memory CD8 T cells based on transcriptomic profiles and were enriched for cytotoxic machinery and immunomodulatory pathways represented by distinct gene co-expression modules. These findings were consistent with complementary evidence from tau mouse models and spinal fluid from AD patients that also implicate CD8 effector memory T cells as the major disease-enriched lymphocyte subtype [25,65]. Lymphocytes were rare in brain tissue; therefore, these observations were remarkable for their consistency across 70 donors (ranging from 5 to 26 cells per donor) and group-level statistical analyses.

Another major question is how lymphocytes interact with different classes and subclasses of brain cells and their disease-associated cell states including microglia. Therefore, we annotated different states of disease-associated microglia to consider their relationships with other brain cell types. Unlike cell classes and subclasses which can be robustly mapped to reference atlases, microglia disease states are highly variable across individuals and studies and lack a consensus reference. To enhance the identification of stable microglial disease states, we leveraged hdWGCNA to identify gene co-expression network modules independent of dimension reduction methods and unsupervised clustering parameters. The resulting microglial modules from human brain data captured reproducible states with distinct pathological associations and markers that have been validated in prior studies [9,10]. This included homeostatic microglia [41] in addition to two distinct microglia modules with differential relationships to tissue accumulation of amyloid and tau pathology, including microglia correlated with beta-amyloid accumulation that are depleted in *APOE4* carriers (ARM; *CD163*+) [9] and microglia related to the accumulation of tau-pathology and neurodegeneration (*GPNMB*+) [66]. In contrast, microglia bearing markers of dystrophic morphology (*FTL*+) [9,67] did not accumulate with pathology or dementia status. These modules also reproduced compositional relationships with different AD pathological stages and *APOE4* status, further supporting their interpretation as robust states.

Using cell–cell interaction analysis, we observed a marked increase in the number of significant CCIs in the disease brains compared to controls. Importantly, the majority of the top 50 gained CCIs involved neuronal subtypes that were significantly depleted from AD dementia tissues, including L2/3 IT excitatory neurons and Sst interneurons. Therefore, extensive changes in intercellular cross-talk may participate in the fate of vulnerable neuronal subclasses in disease, which requires further study. This finding is consistent with the emerging notion that cellular vulnerability likely involves non-cell autonomous factors [27,68].

Previous studies have demonstrated that microglia in AD pathogenesis likely have multifaceted roles, including both immunosuppressive/neuroprotective and inflammatory/neurotoxic [13,16,17,18,19,20]. In our analysis, ARM showed the highest cell–cell interactions of any other cell type, including multiple neuroprotective and glial protective interactions such as prosaposin (*PSAP*) paired with expression of *GPR37L1*, which is a known mediator of anti-oxidant effects in astrocytes [60]. The negative correlation with *APOE4* status further supports likely beneficial roles for ARM in disease. On the contrary, CCI analysis linked pathology-associated microglia with cellular damage responses. In conclusion, our analyses support dual roles for microglia whose interactions with vulnerable neuronal populations included both neuroprotective and neurotoxic pathways that varied according to microglia disease states as annotated by robust marker genes.

Here, we reported an increased number of CD8 T lymphocytes in AD brain tissue samples based on the snRNAseq of 84 cases. The role of lymphocytes in the AD brain was previously considered quite controversial and more recently brought to greater attention by highly sensitive quantitative techniques including cell-specific analyses [25,26]. Therefore, much remains unknown in this rapidly developing field, supporting the importance of the convergence over effector memory T cells as a major cell type observed here in the human brain and previously reported in human AD CSF and tau mouse models [24,25,69]. In combination, this provides a line of independent evidence to elevate the importance of studying these cells more in the context of AD. Emerging evidence suggests that CD8 T cells in AD brains may infiltrate into the brain from systemic circulation, possibly via the *CXCL16–CXCR6* axis as previously reported from the analysis of patient-derived CSF lymphocytes [24,69]. There is also growing awareness of lymphocyte populations that reside in the meninges that may play specialized roles in brain tissue surveillance or as yet undefined roles [70,71]. As new data emerge, evidence builds to support a need for basic and human-specific study into the mechanisms and functions of lymphocytes in the AD brain.

Our findings indicate that CD8 T cells likely have multiple roles in AD pathogenesis, including the potential for extensive immunomodulatory and cytotoxic effects involving multiple cell types, including ARM. Future work, including longitudinal studies, are needed to clarify the relationship between CD8 T cells and AD neuropathology, brain atrophy, and clinical decline. As our results from the human brain overlap with findings previously reported in spinal fluid studies [24,65], it may be possible to use spinal fluid from AD clinical cohorts to probe these associations in future studies. There is also interest in identifying distinct markers of AD-associated lymphocytes to support their identification and future studies. Among the few genes we found to be differentially expressed by CD8 T cells in disease samples, *ERC1* is a regulatory component of *IKKB*, which is part of the NFkB canonical pathway regulation that is typically induced by proinflammatory ligands such as TNF, IL-1, or TLR ligands [46,72]. High mitochondrial and ribosomal RNA in the differentially expressed genes could be indicative of low-quality cells as conventionally interpreted; however, recent single-cell multiomic studies have shown the presence of transcription erosion in severely affected cases [27,73], which also aligns with the terminal differentiation of the enriched CD8 T cell population. Our findings support the value of differential expression analysis of lymphocytes in AD brain tissue; however, larger numbers of lymphocytes or more sensitive techniques are needed to probe these relationships further.

We acknowledge several additional technical limitations of this study. Principally, brain tissue-associated lymphocytes are also not necessarily located within the brain parenchyma, and cells we designated as microglia throughout the analysis may include perivascular macrophages. We omitted cases with neocortical Lewy body pathology concurrent with AD pathology in order to focus our analysis. Moreover, the limited number of non-AD samples with Lewy body pathology in SEA-AD, along with confounders with age, precluded the extension of our analysis to determine the effects of co-pathology on the neuroimmune axis. Finally, our results are observational and do not prove causal or functional associations.

## 5. Conclusions

An in-depth examination of the SEA-AD snRNAseq dataset focusing on CD8 T cells and microglia unveiled a significant presence of CD8 T cells escalating with pathological progression. Notably, we identified prominent cell–cell interactions (CCIs) involving CD8 T cells and amyloid-responsive microglia. Within the expanding literature, our analysis reinforces CD8 T cells as a noteworthy cell type of interest in Alzheimer’s disease (AD). It also identifies candidate molecular and cellular interactions, contributing new insight to our rapidly advancing understanding of the potential role of these cells in AD.

## Figures and Tables

**Figure 1 biomedicines-12-00308-f001:**
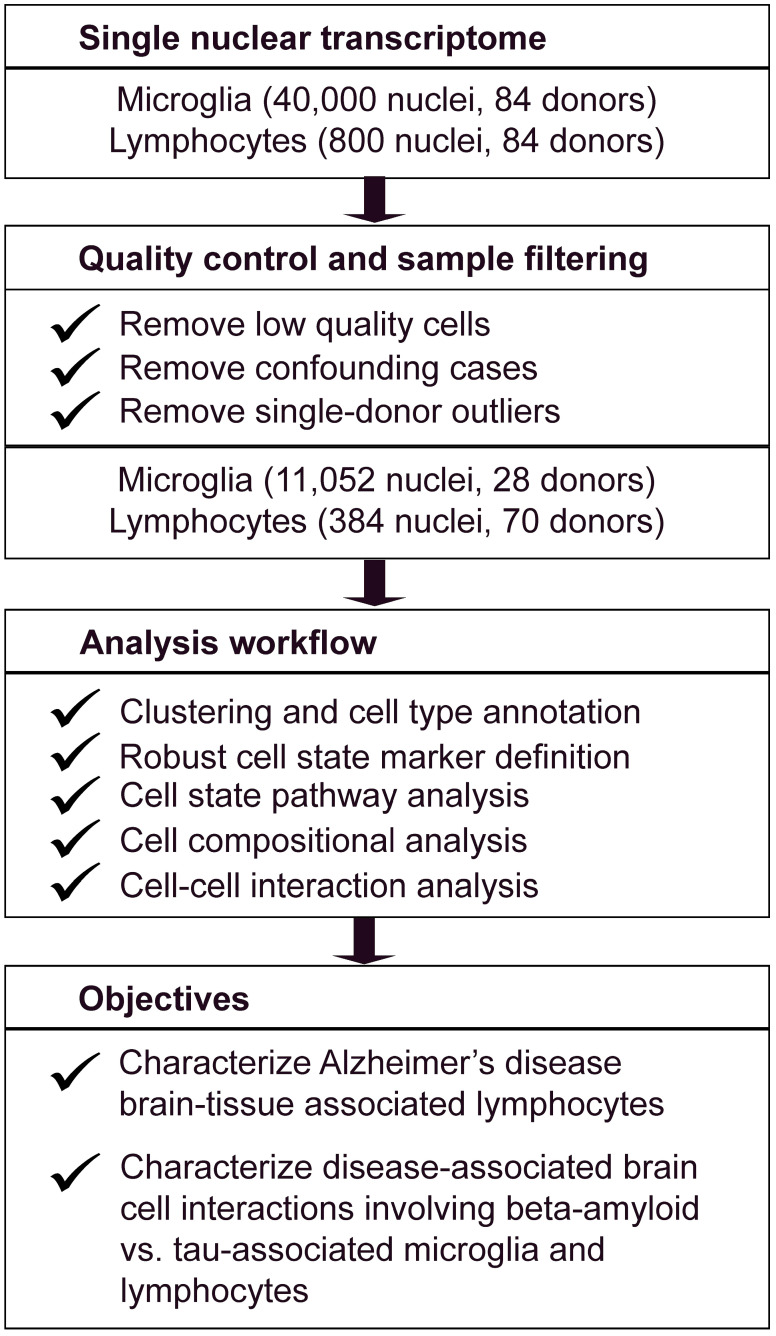
Schema of analysis workflow.

**Figure 2 biomedicines-12-00308-f002:**
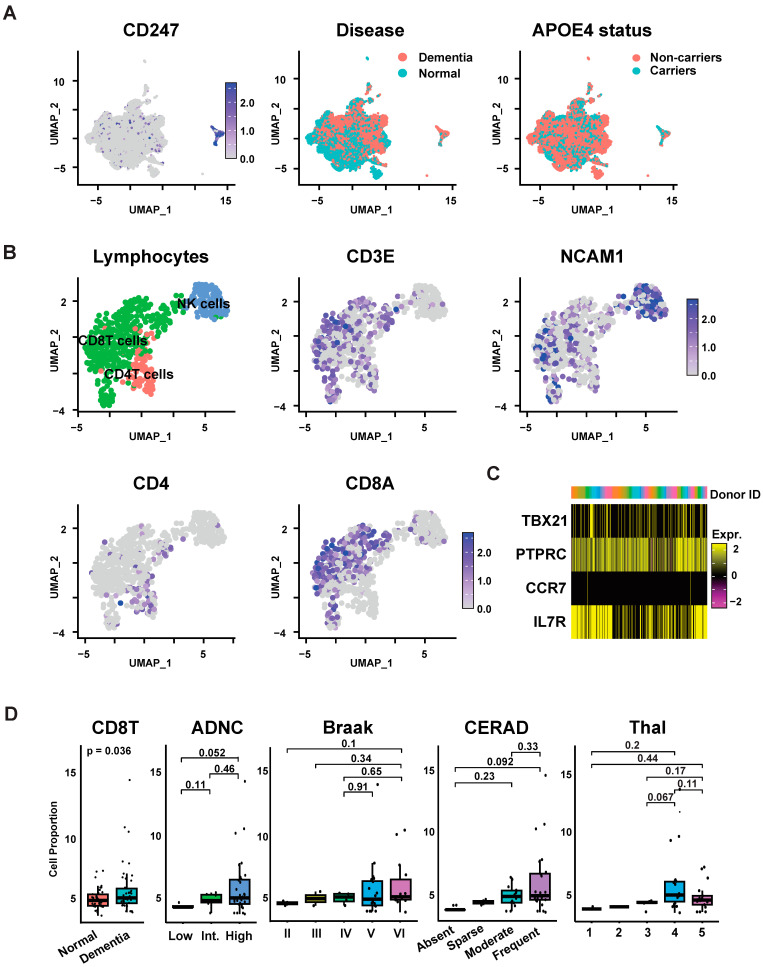
CD8 T cells are abundant in AD brains. (**A**) left—Distinct subcluster of *CD247*+ cells identified in microglia-perivascular macrophages (PVM) Seurat object downloaded from Seattle Alzheimer’s Disease Brain Cell Atlas. Middle, right—UMAP plot of microglia and lymphocytes showed reasonable intermixture of cognitive status (disease) and *APOE4* carrier status. (**B**) left—UMAP plot representing major lymphocyte subtypes in the *CD247*+ cluster, identifying independent subclusters of CD8 T cells, CD4 T cells, and natural killer (NK) cells. Middle and right—Feature plots represent normalized expression of *CD3E*, *NCAM1*, *CD4*, and *CD8A* based on Seurat. (**C**) Heatmap of scaled expression of the representative markers for effector memory T cells with re-expression of CD45RA (PTPRC). *X*-axis color bar shows that none of the clusters are comprised of a single donor. (**D**) Box plot of cell proportion of CD8 T cells. Cell proportions were calculated as number of CD8 T cells per sample divided by the total cell number per sample in the microglia-PVM object. Cell proportions and the *p*-value are from a regression model including cognitive status, age, sex, and postmortem interval (PMI); *Y*-axis represents estimate from the model. For pathological staging (ADNC, CERAD, Braak, Thal), only dementia cases are plotted.

**Figure 3 biomedicines-12-00308-f003:**
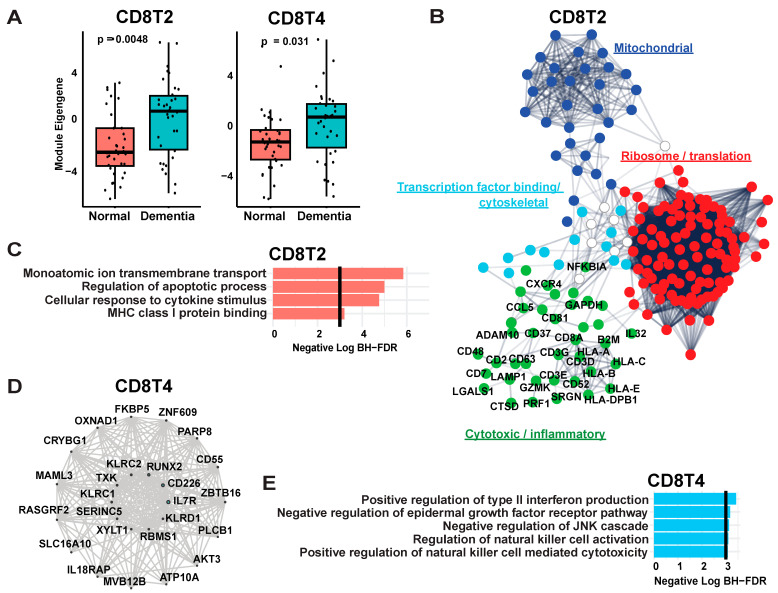
Gene co-expression network modules of CD8 T cells in AD brain. (**A**) Box plot of mean module eigengenes per sample between dementia and control cases for cytotoxic (CD8T2) and immunomodulatory (CD8T4) modules. *Y*-axis represents estimate, derived along with *p*-values from the linear regression model including disease, age, sex, and PMI. (**B**) Pathway enrichment and protein-protein interactions in CD8T2 module. The network plot denotes proteins with CD8T2 kME > 0.2, denoting ribosome/translation pathway, mitochondrial pathway, and cytotoxic/inflammatory pathway. (**C**) The bar plot shows representative significant GO biological process enrichment terms of genes with kME > 0.2. *X* axis shows negative log of Benjamin-Hochberg-corrected *p*-values. (**D**) Network plot and pathway enrichment in CD8T4 module. Inner 10 genes of the network plot are top 10 hub genes for CD8T4 module, and the point size denotes kME. (**E**) The bar plots show representative significant GO biological process enrichment terms of genes with kME > 0.2. *X* axis shows negative log of Benjamin-Hochberg-corrected *p*-values.

**Figure 4 biomedicines-12-00308-f004:**
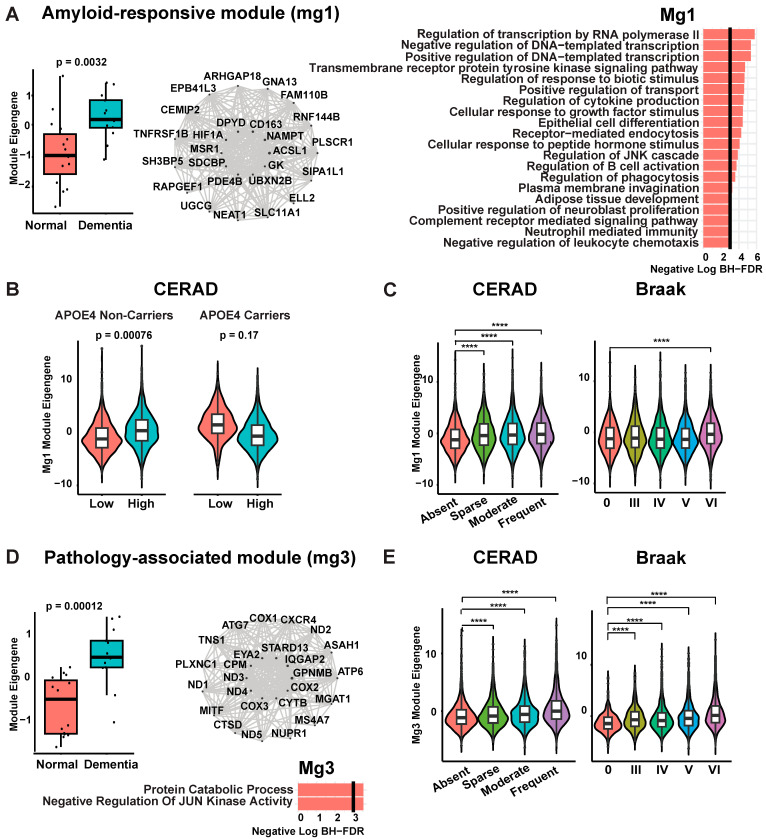
Gene co-expression network modules of microglia in AD brain. (**A**) Amyloid-responsive module (mg1). Top left—box plot of mean module eigengenes per sample between dementia and control cases. *Y*-axis represents estimate, derived along with *p*-values from the linear regression model including disease, age, sex, and PMI. Top middle—network plot of top 20 hub genes for mg1 module. Inner 10 genes of the network plot are top 10 hub genes for mg1 module, and the point size denotes kME. Top right—the bar plots show representative significant GO enrichment terms of genes with kME > 0.2. *X* axis shows negative log of Benjamin-Hochberg-corrected *p*-values. (**B**) Violin plot with overlaying boxplot showing per-nucleus module eigengene of mg1 across senile plaque burden, partitioned to *APOE4* carriers and non-carriers. CERAD score was binned for low (absent and sparse) and high (moderate and frequent). *Y*-axis shows estimates and *p*-values derived from a mixed random effect model including pathological stages, age, sex, PMI; and Donor ID as the random effect. (**C**) Violin plot with overlaying box plot showing per nucleus module eigengene for mg1 module across pathological stages for beta-amyloid (CERAD score) and phosphorylated-tau (Braak stage). *Y*-axis shows estimates and *p*-values derived from a mixed random effect model including pathological stages, age, sex, PMI; and Donor ID as the random effect. ****; *p* < 0.0001. (**D**) Pathology-associated module (mg3). Left—box plot of mean module eigengenes per sample between dementia and control cases. *Y*-axis represents estimate, derived along with *p*-values from the linear regression model including disease, age, sex, PMI. Right top—network plot of top 20 hub genes for mg1 module. Inner 10 genes of the network plot are top 10 hub genes for mg3 module, and the point size denotes kME. Right bottom—the bar plots show representative significant GO biological process enrichment terms of genes with kME > 0.2. *X*-axis shows negative log of Benjamin-Hochberg-corrected *p*-values. (**E**) Violin plot with overlaying box plot showing per nucleus module eigengene for mg3 module across pathological stages for beta-amyloid (CERAD score) and phosphorylated-tau (Braak stage). *Y*-axis shows estimates and *p*-values derived from a mixed random effect model including pathological stages, age, sex, PMI; and Donor ID as the random effect. ****; *p* < 0.0001.

**Figure 5 biomedicines-12-00308-f005:**
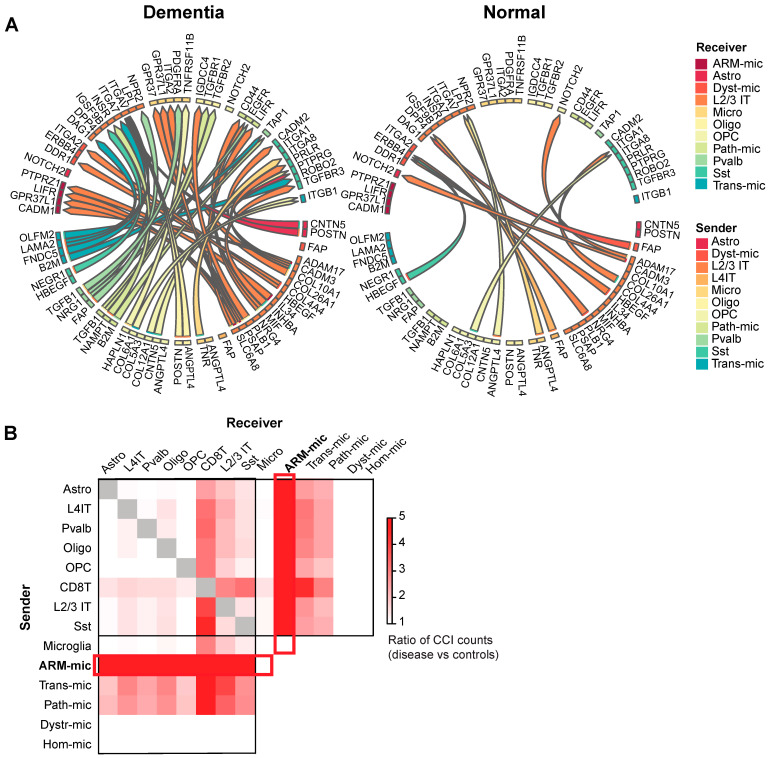
Cell-cell interaction analysis of AD brains. (**A**) Circos plot of top 50 ligand and receptor (LR) pairs. Left—dementia specific, right—normal cases specific interactions. (**B**) Heatmap showing the ratio of sums of prioritization scores among total LR pairs between dementia and normal cases. *X*-axis lists receiver cell types, and *Y*-axis lists sender cell types. Grey cells denote autocrine signals as these were removed from the analysis. AMR-mic—amyloid responsive microglia; Astro—astrocytes; Dyst-mic—dystrophic microglia; L2/3 IT or L4 IT—layer 2/3 or layer 4 intratelencephalic excitatory neurons; Oligo—oligodendrocytes; OPC—oligodendrocytic progenitor cells; Path-mic—pathology-associated microglia; Pvalb—palvalbumin neurons; Sst—somatostatin neurons; Trans-mic—transition microglia.

**Figure 6 biomedicines-12-00308-f006:**
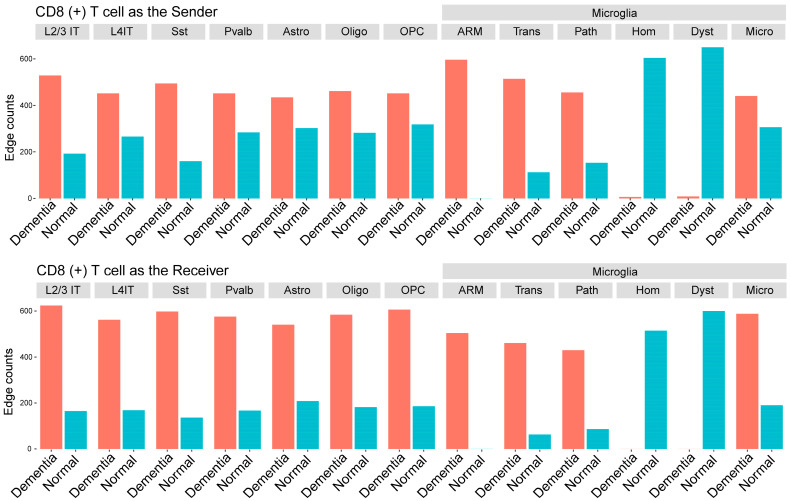
Cell-cell interactions involving CD8 T cells. Bar plots showing the number of cell-cell interactions inferred by MultiNicheNet. (**Top**)—CD8 T cells as the sender, (**bottom**)—CD8 T cells as the receiver. Each panel represents the corresponding interaction partners. ‘Micro’ represents microglia without specific cell state labels such as Homeostatic (Hom), Dystrophic (Dyst), ARM, Transition (Trans), and Pathology-associated (Path).

## Data Availability

All data used in this study are publicly available at CELLxGENE (https://cellxgene.cziscience.com/collections/1ca90a2d-2943-483d-b678-b809bf464c30, (accessed on 23 February 2023)).

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
