# Peer review of "Cell States and Interactions of CD8 T Cells and Disease-Enriched Microglia in Human Brains with Alzheimer’s Disease"

_biomedicines, 2024, doi:10.3390/biomedicines12020308_

Round 1

Reviewer 1 Report

Comments and Suggestions for Authors

The paper is a groundbreaking exploration into the cellular dynamics within Alzheimer's-affected brains. However, the technologies used are so arcane that they might be impossible to grasp for clinicians, so a bit of explanation might shed some light on the methods.

  1. What is the main question addressed by the research?

Whether lymphocytes in the peripheral circulation (which should be separated from the brain by the BBB) have any interaction relevant to the AD with microglia in the brain, which are the equivalent of the immune cells in the brain.

2. Do you consider the topic original or relevant in the field? Does it

address a specific gap in the field?

It is absolutely relevant, as it is addresses a previously unanswered question, of the communication between the immune systems of the body and the brain and whole system, and it demonstrates that we might be able to identify peripheral cells that might be markers for the AD, much easier for the diagnosis.

3. What does it add to the subject area compared with other published

material?

The extraordinary technological approach and the use of high productivity and specificity methods ,that I have never seen used in such a manner, It is highly revolutionary and impressive

4. What specific improvements should the authors consider regarding the

methodology? What further controls should be considered?

The method should be expanded to actual patients, and correlated perhaps with imaging investigations. I have no doubt that such experiments are already underway. The identification of those CD8 lymphocytes in conjunction with high-yield imaging measurements or perhaps isotopic functional measurements could be of major importance

5. Are the conclusions consistent with the evidence and arguments presented

and do they address the main question posed?

Yes

6. Are the references appropriate?

Yes

Reviewer 2 Report

Comments and Suggestions for Authors

This is a complex paper in which the authors performed multiple in silico analyses on a database of Alzheimer disease (AD) brains and non-demented controls in which single nuclear RNA sequencing (snRNAseq) was carried out on multiple tissue samples. These samples are part of a larger brain collection in Seattle, WA.

The results were pre-sorted into different cellular types, based on gene expressions that defined each type, and the authors then sought information on :1) the relative population of each cell type; and 2) cell-cell interactions (CCI) among the different cell types. The authors found that AD temporal gyrus samples had increased numbers of lymphocytes bearing CD8 properties, and that these CD8 lymphocytes had CCI notably with microglia.

Increased neuroinflammation is increasingly being reported as a potential causative agent in degenerative brain diseases in addition to other conditions. Neuroinflammation also appears to be linked causally to increased oxidative stress that has been described extensively in degenerative brain conditions like AD. The importance of this study is that: 1) it confirms in an indirect manner that AD cases are heterogenous, and that no one single "cause" is likely to be discovered; and 2) complicated studies of multiple cell types in brain tissue (like the present study) will be required to define the problems in AD and develop targeted therapies.

Thus, I am in favor of the overall approach taken by the authors. However, my enthusiasm is reduced for the following reasons:

1. The manuscript is basically unreadable except by persons expert in the algorithms used by the authors. The authors provide an Introduction that is excellent and should be retained. The Methods can be presented in very condensed form, and in detail either as Supplementary Methods, or better, embedded in each Supplemental Figure legend.

2. It is unclear what each algorithmic approach to the dataset yielded. These outcomes need to be presented in simple language, so that the reader learned what was found and what it might mean. The major findings of this study get buried in the verbage describing how the multiple algorithms were applied.

3. The authors excluded samples with large number of Lewy bodies (LB). As the authors know, LB's contain alpha-synuclein among other proteins, are found both in non-demented controls and those with "LB dementia", and may themselves be triggers of inflammation. While I understand the rationale behind this choice, it is unclear whether this LB-burdened population was "different" immunologically compared to the main populations. Also, what metric was used to decide the level of LB formation that was "too elevated".

Comments on the Quality of English Language

English is fine except for a very few misspellings and subject-verb mismatches. These few mistakes can easily be corrected with careful editing.

Reviewer 3 Report

Comments and Suggestions for Authors

My suggestions:

1, Some tables and figures may be used in the manuscript itself, rather than in the supplement. The authors missed the description of Supplementary Figure 1.  in the main text. 

2. In the Introduction or Discussion, authors may mention the dual role of microglia in AD (even though it may impact amyloid/Tau clearance, microglial dysfunctions may impact neurodegeneration)

3. In the Methods section, I suggest a workflow figure for the experiments.

4. Authors may discuss more in detail how T cells reach the brain in AD and dementia patients.

5. Authors may discuss the most significant differentially expressed genes in CD8 T cells in the discussion in a little more detail. 

6. I could not find the supplementary tables in the suppelement files. 

Round 2

Reviewer 2 Report

Comments and Suggestions for Authors

This is the first revision of a paper I previously reviewed. The authors have undertaken major changes (deletions/additions) to the original manuscript, and the revision is more readable and accessible. It is still a complex study, and it is unclear whether the filtered data set was used for all calculations except the population of CD8 lymphocytes or not. This is a minor point which is easily dealt with.

If I were publishing these results, I would publish two papers and put the data on cell-cell interactions (CCI) in the second paper. However, I do appreciate the authors' desire to have everything in one longer (and more complex) paper.

Also, I'm curious in light of their findings whether the authors feel that the CD8 T cell population could be detected in blood samples of living AD subjects? This type of study is obviously not feasible in post-mortem studies. Also, they refer to the mutant tau mice, but I'm curious if the many beta amyloid mutant mice treated with anti-amyloid Ab's have any changes in amyloid-reactive microglia? CD8 T cells? CCI? This is an approachable experiment, and I hope the authors might consider such a collaboration in the future.

In any event, their revised paper is much improved over the original and demonstrates the power of a cell nuclei-based gene expression analysis using recently developed algorithms. Support its publication as is.

Reviewer 3 Report

Comments and Suggestions for Authors

The authors fulfilled my suggestions. Thank you.